



# Improving scalability of Earth System Models through coarse-grained component concurrency - a case study with the ICON v2.6.5 modeling system

Leonidas Linardakis[1], Irene Stemmler[2], Moritz Hanke[3], Lennart Ramme[1], Fatemeh Chegini[1], Tatiana Ilyina[1], and Peter Korn[1]

[1]Max Planck Institute for Meteorology, Hamburg, Germany
[2]wobe-systems GmbH, Kiel, Germany; previously at the Max Planck Institute for Meteorology
[3]Deutsches Klimarechenzentrum, Hamburg, Germany

**Correspondence:** L. Linardakis (leonidas.linardakis@mpimet.mpg.de)

**Abstract.**

In the era of exascale computing, machines with unprecedented computing power are available. Making efficient use of these massively parallel machines, with millions of cores, presents a new challenge. Multi-level and multi-dimensional parallelism will be needed to meet this challenge.

Coarse-grained component concurrency provides an additional parallelism dimension, that complements typically used parallelization methods such as domain-decomposition and loop level shared memory approaches. The novel aspect is that component concurrency is a function parallel technique, while these parallelization methods are data parallel techniques. This additional dimension of parallelism allows us to extend scalability beyond the limits set by the established parallelization techniques. Furthermore, concurrency allows each component to run on different hardware, and thus leveraging the usage of heterogeneous hardware configurations.

We study the characteristics of component concurrency and analyse its behaviour in a general context. These generic considerations are complemented by an analysis of a specific case, namely the coarse-grained concurrency in the multi-level parallelism context of two components of the ICON modeling system: the ICON ocean model ICON-O and the marine biogeochemistry model HAMOCC. The additional computational cost incurred by the biogeochemistry module is about three times that of the ICON-O ocean stand alone model, and traditional parallelization techniques present a scaling limit that impedes the computational performance of the combined ICON-O-HAMOCC model. Scaling experiments, with and without concurrency, show that component concurrency extends the scaling, in cases doubling the parallel efficiency. The experiments' scaling results are in agreement with the theoretical analysis.

## 1   Introduction

Since the dawn of modern computing, numerical weather prediction and climate modeling have been among the first scientific applications to make use of the new technology (Dalmedico (2001); Washington et al. (2009); Balaji (2013)). In the decades





following the creation of the first atmosphere computer models the computational power has been increasing exponentially (McGuffie and Henderson-Sellers (2001)), allowing the development of complex Earth System Models (Randall et al. (2018)) running at ever higher resolutions. A first impressive result was described by Miyamoto et al. (2013), where the atmosphere model NICAM ran in a global sub-kilometer resolution for 12 simulated hours, dynamically resolving convection. Since then, other groups have followed the path of reducing parameterizations by increasing the resolution, as for example in global storm resolving setups described in Stevens et al. (2019). We are currently viewing the perspective of constructing the Earth's digital twin (Voosen (2020); Bauer et al. (2021)), where much of the Earth's system complexity will be captured by models at one kilometer resolution.

These developments have been made possible by the availability of massively parallel computers. Since the beginning of the 21st century the focus of CPU development has switched from constructing more powerful processing units to packing more units into an integrated chip. In response, programmers had to turn much of their efforts from optimizing the code to efficiently parallelizing it (Sutter (2005); Mattson et al. (2008)). The era of exascale computing is here with the construction of machines like the Frontier at the Oak Ridge National Laboratory. While less than fifteen years ago we were facing the challenge of petascale computing (Washington et al. (2009)), we are now facing a new level of challenge: how to efficiently parallelize our codes for machines with millions of cores.

The parallelization backbone of Earth system models consists of domain decomposition techniques, where the horizontal grid is decomposed into subdomains, which are assigned to different processing units, and the Message Passing Interface (MPI, Walker (1992); The MPI Forum (1993)) is used to communicate information between them. This approach has been designed primarily for distributed memory parallelization. In the past years it has been apparent that domain decomposition methods alone cannot efficiently scale when using high number of cores placed on a shared memory board. Since two decades, shared memory parallelization mechanisms, such as OpenMP (Mattson (2003)), have being developed. These have been increasingly employed for providing loop-level shared memory parallelization, in order to exploit the new multi-core architectures. More recently, GPUs have attracted a lot of attention due to the high computing power they provide through massive parallelism, while at the same time require lower power consumption per FLOP (floating point operations per second) than traditional CPUs. The two levels of parallelization that are currently widely used, domain decomposition with MPI for distributed memory parallelization, and OpenMP shared memory loop-level parallelization (or similar approaches, like OpenACC for GPUs), have so far been successful in yielding satisfactory performance on parallel machines. They still pose though some limitations, as their scaling efficiency typically depends on the amount of grid points available for parallelization.

The concept of concurrency goes back to before parallel computing came into practice (Lamport (2015)). It refers to algorithmic dependencies and independences, and was first developed in the context of multitasking. The term has come to be synonymous to task parallelism, as independent tasks can run in parallel. In contrast to the domain decomposition and loop-level parallelization methods, concurrency is a function (or task) parallel approach. Coarse-grained component concurrency is a special case of concurrency, where the independent components are large model modules, essentially sub-models, with comparable computational workload. It has been used in climate modeling since decades in atmosphere-ocean coupled setups: the two models run in parallel, and are coupled every one or more time steps. More recently, the same idea has been applied





on the radiation component of the atmosphere (Mozdzynski and Morcrette (2014); Balaji et al. (2016)). These results have shown promise that component concurrency can be applied to other modules of the models, and that the technique can provide leverage for running on many-core architectures.

Increasing the grid resolution allows us to resolve smaller scales, to better approximate the physical processes, and to rely less on parameterizing unresolved processes. This increased problem size can still be effectively parallelized using data parallelism, at least up to a point. On the other hand, there is interest to include more processes into Earth system models, in order to have a more detailed representation of the Earth system. Such processes may represent the atmosphere chemistry, the cryosphere and the ocean biogeochemistry, and can have a significant impact on the Earth's climate and the biosphere.

Ocean biogeochemistry comprises a variety of chemical and biological processes in the water column and the sediments of the ocean (Sarmiento and Gruber (2006)). These processes include, for example, biological activity of phytoplankton, zooplankton and different types of bacteria, the chemical and biological cycles of carbon or nitrogen, and the dissolution of gases in seawater. Ocean biogeochemistry is therefore an important component for quantifying critical developments in the Earth system, like the oceanic uptake of anthropogenic $CO_2$ released from fossil fuels (Ciais et al. (2014)), or ocean

acidification and deoxygenation under the impact of global warming (Orr et al. (2005); Breitburg et al. (2018)). However, the number of processes that could be included is extensive, and ocean biogeochemistry models are becoming ever more complex through the addition of more tracers and processes (Ilyina et al. (2013)). This results in a large computational cost, which hinders their integration in high resolution Earth system models, especially when simulating the long time scales that are crucial to investigate changes in the biogeochemical state of the ocean.

In contrast to increasing the grid size, the additional computational cost imposed by introducing new processes cannot be absorbed through grid decomposing parallel methods, as these are limited by the grid size. Component concurrency offers a way to increase the model complexity, while maintaining reasonable performance. Not all components though may be subjected to concurrency in the same way, or as effectively. Side effects, like instabilities and change of the model behavior, may emerge; Balaji et al. (2016) offer a first discussion of such issues.

In this paper we study the impact of component concurrency on the scaling behavior of a model. We examine it in a general abstract context, in what manner component concurrency differs from the more traditional approaches, and what its scaling characteristics are. We consider concurrency to be part of a multi-level parallelism scheme, and we examine the cases where it can improve performance, and when this improvement is optimal. The ICON-O-HAMOCC ocean biogeochemistry model offers a good test-case, as it is about four times slower than ICON-O alone. We have run two sets of experiments on two

different machines. The scaling results obtained from the experiments are in good agreement with the predictions from the theoretical analysis.

In Section 2 we give a brief description of the ICON models. In Section 3 we examine the behavior of component concurrency in a general context. Section 4 describes the basic steps to engineer concurrency for ICON-O-HAMOCC. Experiments and the results are presented in Section 5. A comparison of the experimental results with the theoretical analysis is given here.

In Section 6 we give an overview of the results and future work.



## 2 ICON model description

ICON is an Earth system model framework developed in collaboration with the German Weather Service (DWD), the Max
Planck Institute for Meteorology (MPIM), the Institute of Meteorology and Climate Research at the Karlsruhe Institute of
Technology, and the German Climate Computing Centre (DKRZ). It consists of the numerical weather prediction model ICON-
NWP (Zängl et al. (2015)), the climate atmosphere model ICON-A (Giorgetta et al. (2018); Crueger et al. (2018)) and the land
model JSBACH (Nabel et al. (2020)), the ocean model ICON-O (Korn et al. (2022)), the atmosphere aerosol and chemistry
model ICON-ART (Rieger et al. (2015)), and the marine biogeochemistry model HAMOCC (Ilyina et al. (2013)). The ICON
Earth system model ICON-ESM consists of ICON-A, JSBACH, ICON-O and HAMOCC (Jungclaus et al. (2022)).

The ICON horizontal grid consists of triangular cells constructed by recursively dividing the icosahedron (Tomita et al.
(2001)), and it provides near-uniform resolution on the sphere. More general non-uniform triangular grids can be used by
ICON-O (Logemann et al. (2021)).

The ICON framework provides common infrastructure to its components. It supplies the domain decomposition routines,
and model contextual high level communication interfaces to the Message Passing Interface (MPI). It also provides flexible
interfaces for input, and automatic parallel asynchronous output mechanisms. The YAC library (Hanke et al. (2016)) serves
as a general coupler between models. Models are registered to a simple master control module through namelists. The ICON
models employ both domain decomposition and OpenMP loop level parallelism. ICON-A and JSBACH can also run on GPUs
using OpenACC directives (Giorgetta et al. (2022)).

### 2.1 The ICON-O ocean model

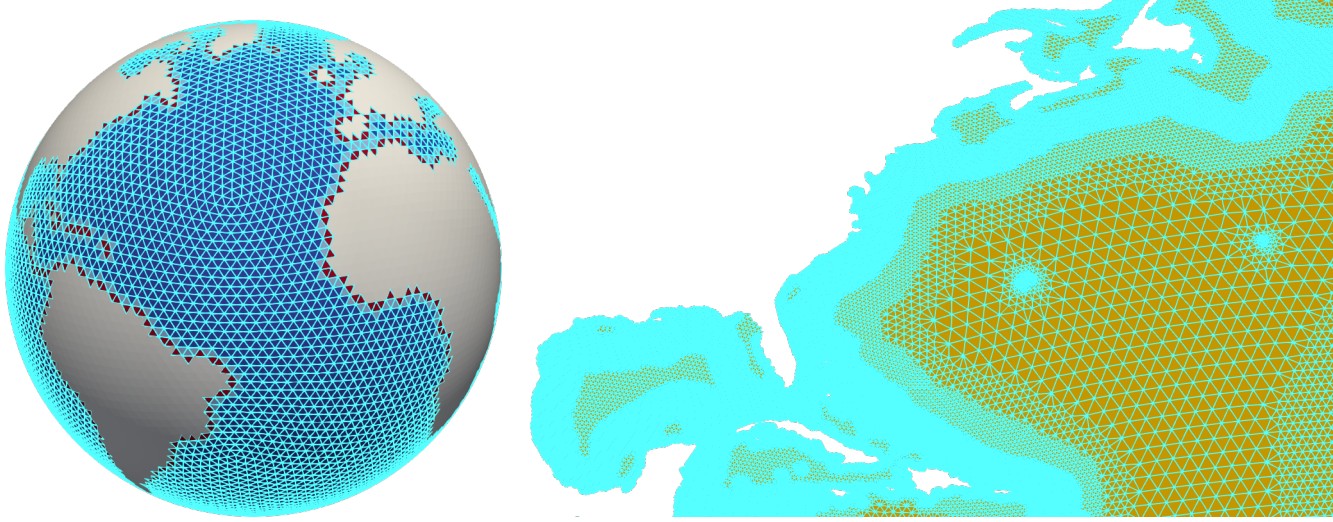

**Figure 1.** Left: The uniform ocean icosahedron-based grid at 160km resolution, the sea-land mask is in color. Right: Detail from the non-
uniform global coastal ocean grid with resolution 8km–80km used in Mathis et al. (2021).





ICON-O is the ocean general circulation model that provides the ocean component to the ICON-ESM. Its horizontal spatial
discretization is based on unstructured triangular grids, allowing a variety of setups, from idealized basins (Korn and Danilov (2017)), to global ocean domains, where the interior land points are removed, see Fig. 1 left. In uniform setups ICON-O has been tested in global 1km resolutions, Hohenegger et al. (2022). In non-uniform setups it has been tested with "telescoping" setups (Korn et al. (2022)), which can produce locally resolutions to 600m (Hohenegger et al. (2022)); this level of refinement allows to locally resolve submesoscale features. Furthermore, a topographic and coastal adaptive local refinement (Logemann et al. (2021)) has been used for global coastal ocean simulations (see Fig. 1 right). This setup includes the HAMOCC biogeo-chemistry model (see Mathis et al. (2021)).

ICON-O solves the oceanic hydrostatic Boussinesq equations, also referred to as the "primitive equations". The primitive equations are solved on the triangular ICON grid with an Arakawa C-type staggering, using a *mimetic* horizontal discretization, where certain conservation properties of the continuous formulation are inherited to the discretized one. The staggering necessitates reconstructions to connect variables that are located at different grid positions. This is accomplished in ICON-O by utilizing the novel concept of *Hilbert space admissible reconstructions*, for details see Korn (2017), Korn and Linardakis (2018).

The vertical coordinate axis is given by the z-coordinate, which reflects the geopotential height. The two-dimensional triangles are extended by a height-based dimension, which generates three-dimensional prisms. Alternative vertical coordinates such as the $z^*$-coordinate are available in ICON-O, they are described in Singh and Korn (2022).

ICON-O is stepping forward in time with a semi-implicit Adams-Bashford-2 scheme. The free surface equation is solved implicitly in time, using an iterative conjugate gradient solver. The remaining state variables are discretized explicitly. For details we refer to Korn (2017).

The subgrid scale closure for velocity uses a biharmonic operator based on the vector Laplacian operator. In non eddy permitting resolutions, the eddy-induced diffusion and eddy-induced advection are parameterized following Redi (1982) and Gent and McWilliams (1990), respectively. The parameterization of vertical turbulent mixing in ICON-O relies on a prognostic equation for turbulent kinetic energy (TKE), and implements the closure suggested by Gaspar et al. (1990). The vertical dissipation and the vertical diffusion are discretized implicitly.

The ICON sea-ice model consists of a dynamic and a thermodynamic component. The sea-ice dynamics follows on the elastic-viscous-plastic (EVP) rheology formulation, and is based on the sea-ice dynamics component of FESIM, see Danilov et al. (2015). The thermodynamics of sea-ice describe the freezing and melting of sea-ice by a single-category, zero-layer formulation (Semtner (1976)).

A combination of a first order upwind scheme and a second order scheme are utilized for the horizontal tracer transport. The second order method utilizes flux calculations by compatible reconstructions, as described in Korn (2017). The two schemes are combined through a Zalesak limiter (Zalesak (1979)), resulting a "flux-corrected transport", which avoids the creation of new extrema (over/undershoots). This combination results in both monotonicity and low numerical diffusion, which are essential for preserving the water density structure.





For the vertical tracer transport we use a combination of the piecewise parabolic method (PPM), see Colella and Woodward (1984), as high-order, and upwind as low order method.

## 2.2 The HAMOCC biogeochemistry model

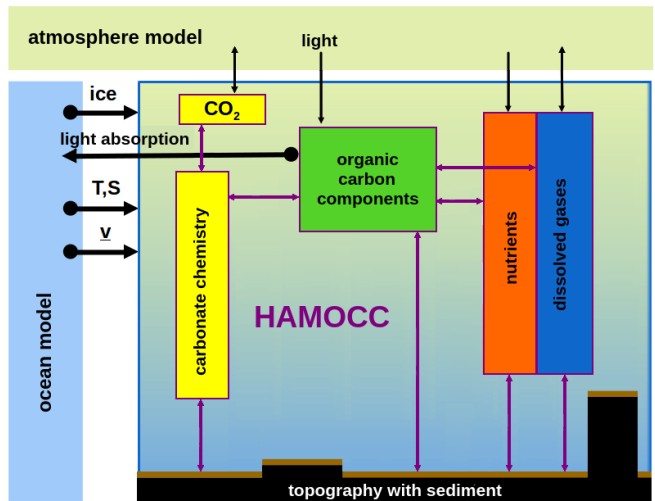

**Figure 2.** Schematic of the biogeochemical processes simulated by HAMOCC. HAMOCC needs to be coupled to an ocean model that provides the fields of temperature, salinity and sea-ice cover, and transports the HAMOCC tracers according to the flow field. The surface inputs can either come from a coupled atmosphere model or are prescribed. The biogeochemistry in HAMOCC can then feed back to the ocean model via the impact of light absorption on temperature and to the atmosphere model via the uptake or release of $CO_2$.

The Hamburg Ocean Carbon Cycle (HAMOCC) model has initially been developed in the earlier work by Maier-Reimer (1984) and Maier-Reimer and Hasselmann (1987) to address the role of ocean processes driving the fate of carbon in the climate system over timescales ranging from seasons to thousands of years. To achieve a consistent evolution of the ocean biogeochemistry, the biogeochemical variables are handled as tracers on the three dimensional grid of the ocean general circulation model. They are transported in the same manner, i.e. using the same numerical methods and time step, as salinity and temperature.

The processes simulated by HAMOCC include biogeochemistry of the water column and upper sediment, as well as interactions with the atmosphere. Figure 2 shows a schematic overview of the key components of the HAMOCC model. In the water column, the biogeochemical tracers undergo modifications by biological and chemical processes, described in detail in Ilyina et al. (2013); Paulsen et al. (2017). At the air-sea interface, the fluxes of $O_2$, $N_2$ and $CO_2$, are calculated. Furthermore, dust and nitrogen deposition from the atmosphere to the ocean are accounted for. The simulation of the oceanic sediment follows the approach of Heinze et al. (1999), and biogeochemical tracers are exchanged with the upper sediment.





Marine biology dynamics connects biogeochemical cycles and trophic levels through the uptake of nutrients and remineralization of organic matter. It is represented by the extended NPZD approach with nutrients, that is dissolved inorganic nitrogen (N), phytoplankton (P), zooplankton (Z), and detritus (D) (sinking particulate matter), and also dissolved organic matter (Six and Maier-Reimer (1996)). Explicit fixation of nitrogen is performed by cyanobacteria (Paulsen et al. (2017)). All organic compounds have identical nutrient and oxygen composition following the Redfield ratio concept extended by a constant ratio for carbon and the micronutrient iron. The treatment of carbon chemistry follows the guide to best practices, as described in Dickson et al. (2007); Dickson (2010).

The transport of biogeochemical tracers presents the most expensive computational part of the HAMOCC model. The number of advected tracers depends on the complexity of the included processes. For example, including organic matter from riverine or terrestrial sources (Lacroix et al. (2021)), extending the nitrogen cycle by including ammonium and nitrite, simulating carbon isotopes or using a more realistic sinking method for particular organic matter (M4AGO scheme: Maerz et al. (2020)), incorporation of stable carbon isotope $^{13}$C (Liu et al. (2021)), increase the number of advected tracers from the default value of 17 and therefore increase the computational cost. Introducing concurrency enables the use of currently simulated processes and may allow the addition of even more tracers, necessary for including more processes, while maintaining an acceptable throughput.

## 3   Coarse-grained component concurrency and multi-dimensional parallelism

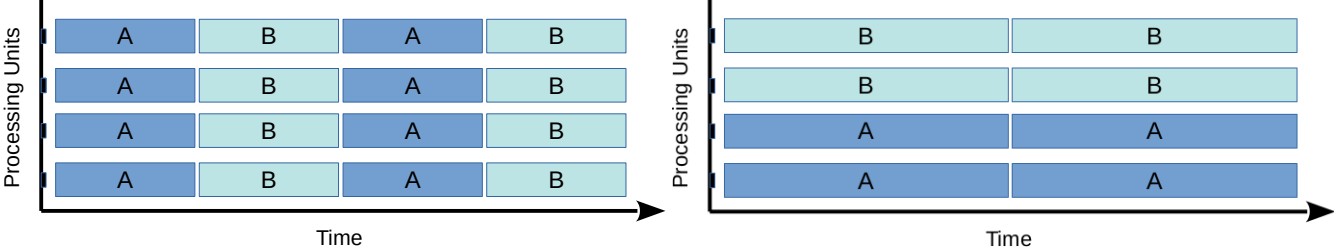

**Figure 3.** Left: two model components $A$ and $B$ running on 4 processing units using only domain decomposition parallelism. Right: the two components run concurrently, each on 2 processing units.

In a coarse component concurrent setup two or more components of the model are run in parallel. The level of "coarseness" is difficult to define, here we will understand it as being able to run concurrently the components throughout a whole timestep. The components are algorithmically independent, and may only need to receive input data from other components. Thus, we expect only one point of communication between the components, where all the information is exchanged. Such components maybe the radiation, the ocean biogeochemistry, the sea-ice, the ice-sheets, etc.





From here on, we will use the term "concurrency", instead of coarse-grained component concurrency, for brevity. We will
also use the term "sequential" as a synonym to "non-concurrent", in the sense that the components run sequential to each other;
other types of parallelization though may still be present.

A schematic of concurrency is drawn in Fig. 3. Let $A$ and $B$ be two components of the model. In the case of using domain
decomposition parallelism only, the domain is decomposed and the subdomains are distributed among the processing units,
while the two modules run sequential to each other, as in Fig. 3 left. In the case of concurrency the two modules run on two
different groups of processing units, depicted in Fig. 3 right.

## 3.1 Levels of parallelism

We can identify three levels of parallelism:

a. *High level* parallelism is applied over the whole model, or over the whole concurrent components of the model. The
most successful such technique is to decompose the horizontal grid, and use MPI for communicating between the subdomain
processes. Component concurrency falls into this category. The most well known example in of concurrency climate modeling
is running the atmosphere and the ocean models concurrently, and coupling them every one or more time steps. An important
characteristic of high level parallelization is that it is independent of the machine architecture. It can be applied across nodes
of heterogeneous machines, and can facilitate hybrid setups, by running simultaneously on different types of processing units.

b. In the *medium level* of parallelism we identify parallel structures on a task or loop level. These are shared memory
parallelization techniques, such as OpenMP or OpenACC. We can consider them to be "medium or fine grained" parallelism.
In contrast to high level parallelization, the implementation of this level of parallelization may not be independent from the
type of architecture.

c. In the *low level* parallelism we identify techniques closer to the architecture, such as vectorization and out-of-order
execution. These techniques depend on the particular architectures, and will not be considered in the following discussion.

Another way to characterize parallelism is by the type of decomposition that is employed. In *data parallelism* the data do-
main is decomposed, and the same operations are applied to each sub-domain. Examples of data parallelism are the domain
decomposition techniques, and the loop level parallelism. In the *function (or task) parallelism*, the algorithmic space is "decom-
posed". Examples are OpenMP task parallelism, out-of-order execution, and also the coarse-grained component concurrency.
So, these two types of decomposition exist across the three levels of parallelism.

Domain decomposition parallelism comes with a communication and synchronization cost, typically caused from exchang-
ing values of "halo" cells between processes. These halo cells consist of the boundary of subdomains which are replicated by
their neighbor subdomains (see Fig. 4). The total number of halo cells generally increases proportional to $\sqrt{N}$, where $N$ is
the number of subdomains. In turn, the parallelization gain for halos is only proportional to $\sqrt{N}$, instead of $N$. This imposes
a limit on how far we can use only domain decomposition as a parallel paradigm for running on massively parallel machines.
We will further examine this behavior in the experiments Section 5.1.



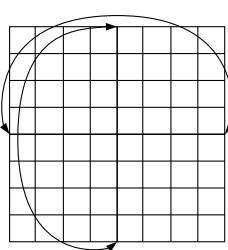
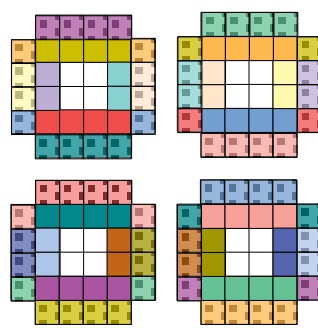

**Figure 4.** Left: a domain of 64 grid points. It is assumed double-periodic for illustration purposes. Right: the domain decomposed in 4 subdomains. Grid points with the same color are duplicated, with halos depicted shadowy.

Another type of communication are global reduction operators, such as global sums. These are typically used in matrix inversions, as is the case for ICON-O, and they also impose scaling limits. We will also observe this behavior in the experiments in Section 5.2.

OpenMP parallelization provides complimentary advantages to the domain decomposition. It offers dynamic load balancing and no communication cost. On the other hand, performance is restricted by latency, memory bandwidth, and in NUMA (non-uniform memory access) machines by data locality. This last disadvantage is alleviated in the domain decomposition approach due to smaller memory footprint per process.

Component concurrency also comes with a cost, which depends on how it is implemented. If we keep the total MPI tasks constant, equal to $N$, we have two options. In the case of a distributed memory parallelization, we split the MPI tasks among the two components, assigning $N_1$ to the first, and $N_2$ to the second, and apply shared memory parallelization inside each component. The cost comes from communicating between the two components.

In the case of a shared memory implementation of concurrency, both components will run on $N$ MPI tasks. In this case we do not have a communication cost, but the synchronization cost remains. Moreover, the performance will partially depend on how efficiently MPI can handle concurrently multiple communicators from the same MPI task. Other aspects may also prove to be significant, like I/O. In the case of distributed memory implementation, all infrastructure is automatically also distributed, including output.

The other aspect of these two options is the software structure. The distributed memory case is a high-level implementation, it is independent of the architecture, and can even be applied in hybrid mode. This is not the case in the shared memory approach. Weighting the pros and cons can only be done in some context. In this work we choose to implement and study component concurrency as a distributed memory approach, due to the high-level parallelism it offers.





## 3.2 Coarse-grained component concurrency and scalability

For climate models the total computing workload is proportional to the grid size[1] and the number of operations per grid point required to solve the problem. We have $W_T = a \cdot s$, where $W_T$ is the total workload, $s$ is the grid size, and $a$ is the number of operations per grid point. We define the *parallel workload* as the workload inside a parallel region, that is between two synchronization points, assigned on one processing unit. For example, the total workload inside an OpenMP parallel loop, divided by the number of OpenMP threads, would constitute a parallel workload. Such a parallel region contains a constant number of operations $a_p$ per grid point[2], so we have

$$W_p = a_p \cdot s/N.$$

Let $A$ and $B$ be two modules of the model using the same grid (as in Fig. 3). Let $W_A$ be the total workload of module $A$ and $W_B = \lambda \cdot W_A$ the total workload of module $B$. Proportionally, let $N_A = N$ be the number of processing units that $A$ runs on its own, and $N_B = \lambda \cdot N$ the additional number of processing units we use when adding module $B$. The total number of processing units is now $N_A + N_B$. Let us consider the parallel workload as the workload assigned to a parallel loop. In the typical data parallel case, where the grid space is decomposed, the parallel workload of a parallel region is $W_p = a_p \frac{s}{N_A + N_B}$, independently if this parallel region belongs to to module $A$ or $B$. In the concurrent case, where module $A$ runs on $N_A$ units and module $B$ on $N_B$ units, the parallel workload of this parallel region is $W_p = a_p \frac{s}{N_A}$, if it belongs to module $A$, and $W_p = a_p \frac{s}{N_B}$ if it belongs to $B$. In both cases concurrency increases the parallel workload compared to data parallelism only. This is a main feature of concurrency compared to data parallelism, it provides another parallelism dimension, by decomposing the function space $a$ instead of the problem size $s$.

How increasing the parallel workload affects the total performance? If we ignore the scaling issues, there is no effect. In the non-concurrent case the time to solution is proportional to $(W_A + W_B)/(N_A + N_B) = ((1 + \lambda) \cdot W_A)/((1 + \lambda) \cdot N) = W_A/N$, while in the concurrent case is $W_A/N = (\lambda \cdot W_A)/(\lambda \cdot N) = W_B/N_B$. The performance is the same. Only when scaling is taken into account, concurrency has an impact. We will examine this impact in the following discussion.

Let $T(1) = r \cdot W_A = r \cdot a \cdot s$ be the time it takes to run module $A$ on one processing unit, where $r$ is a constant that characterizes the computing power of the processing unit, and $W_A$ the workload. We will only consider the homogeneous case, where all units have the same processing power. Let $T(N)$ be the time for running on $N$ processing units. The *speedup* is defined as $S(N) = T(1)/T(N)$, and the parallel *efficiency* as $F(N) = T(1)/N/T(N) = S(N)/N$. We have that

$$T(N) = T(1)/(F(N) \cdot N).$$

Let us now add another component $B$ to the model $A$ as above, that adds a workload of $W_B = \lambda \cdot W_A$, increasing the time cost for running on one processing unit to $T(1) \cdot (1 + \lambda)$. We increase proportionally the number of processing units, from $N$ to $N(1 + \lambda)$. We will assume that our new component $B$ has the same scaling behavior as $A$, so that the same efficiency function

---

[1]The total 3-dimensional grid size.

[2]The number of operations per horizontal grid point may vary depending on conditionals and number of active vertical levels. These differences would create imbalance. Without loss of generality we can take the maximum workload among processes.





$F(N)$ applies also to $B$. When using data decomposition parallelization our new time cost is

$$T_d(N \cdot (1+\lambda)) = \frac{T(1) \cdot (1+\lambda)}{F(N \cdot (1+\lambda)) \cdot N \cdot (1+\lambda)} = \frac{T(1)}{F(N \cdot (1+\lambda)) \cdot N}.$$

When on the other hand we run component $B$ concurrently on $\lambda \cdot N$ nodes, with $A$ on $N$ nodes, then the total time cost is

$$T_c(N \cdot (1+\lambda)) = \max(T_{Ac}, T_{Bc}),$$

where

$$T_{Ac} = \frac{T(1)}{F(N) \cdot N} + C(N \cdot (1+\lambda)),$$

$$T_{Bc} = \frac{T(1) \cdot \lambda}{F(N \cdot \lambda) \cdot \lambda \cdot N} + C(N \cdot (1+\lambda)) = \frac{T(1)}{F(N \cdot \lambda)N} + C(N \cdot (1+\lambda))$$

are the time costs for running components $A$ and $B$ concurrently, and $C(N \cdot (1+\lambda))$ is the cost incurred by the concurrency.

We assume that the scaling is at most linear, that is $F'(N) \leq 0$[3]. Without loss of generality we take $\lambda \leq 1$ (in the opposite case we can just swap the modules $A$ and $B$). Then $F(N \cdot \lambda) \geq F(N)$ and $T_c = T_{Ac}$. Comparing $T_c$ with $T_d$ we have

$$\frac{T_c}{T_d} = \frac{\frac{T(1)}{F(N) \cdot N} + C(N \cdot (1+\lambda))}{\frac{T(1)}{F(N \cdot (1+\lambda)) \cdot N}} = \frac{F(N \cdot (1+\lambda))}{F(N)} + \frac{C(N \cdot (1+\lambda)) \cdot F(N \cdot (1+\lambda)) \cdot N}{T(1)}$$

We set

$$L(N, \lambda) = \frac{F(N \cdot (1+\lambda))}{F(N)},$$

termed *relative efficiency*. We have

$$\frac{T_c}{T_d} = L(N, \lambda) + \frac{C(N \cdot (1+\lambda)) \cdot L(N, \lambda) \cdot F(N) \cdot N}{T(1)}.$$

Taking into account that $T(N) = \frac{T(1)}{F(N) \cdot N}$, we have

$$\frac{T_c}{T_d} = L(N, \lambda) + L(N, \lambda) \cdot \frac{C(N \cdot (1+\lambda))}{T(N)} = L(N, \lambda) \cdot \left(1 + \frac{C(N \cdot (1+\lambda))}{T(N)}\right). \tag{1}$$

We seek the conditions where $\frac{T_c}{T_d}$ is smaller than 1, and as small as possible. In a linear, or near-linear scaling regime, where the efficiency $F(N)$ is nearly constant as a function of $N$, we have $L(N, \lambda) \approx 1$, and concurrency will provide little, if any, benefits.

Let us examine the sub-linear scaling regime. Then $F(N)$ is a strictly decreasing function of $N$, and $L(N, \lambda) < 1$. Moreover $L(N, \lambda)$ is a strictly decreasing function of $\lambda$; when we keep $N$ constant, we have

$$\frac{\partial L(N, \lambda)}{\partial \lambda} = \frac{1}{F(N)} \frac{\partial F(N \cdot (1+\lambda))}{\partial \lambda} < 0.$$

---

[3] We take the liberty to consider $N$ continuous whenever needed.





Concurrency will provide the maximum benefits when $\lambda$ is maximum, that is $\lambda = 1$ (recall that $\lambda \leq 1$), and the two modules
have the same workload. On the other hand, if $\lambda \ll 1$, we have $L(N, \lambda) \approx 1$, and the benefits would be significantly reduced.
In this case the bulk of the workload is beared by module $A$, and the additional parallelism for $B$ provides little profits. The
"coarse-grained" part of the concurrency does not hold, and one should consider to use fine grained parallelism.

The sub-linear scaling property is not sufficient to allow us deduce the behavior of $L(N, \lambda)$ as a function of $N$. We will
further accept that the scaling behavior follows Amdahl's Law (Amdahl (1967)). In this case $T(N) = T(1) \frac{(1-\sigma) + \sigma \cdot N}{N}$, where
$0 < \sigma < 1$ is the part of the code that does not scale. Then $S(N) = \frac{T(1)}{T(N)} = \frac{N}{(1-\sigma) + \sigma \cdot N}$, $F(N) = \frac{S(N)}{N} = \frac{1}{(1-\sigma) + \sigma \cdot N}$, and
$L(N, \lambda) = \frac{(1-\sigma) + \sigma \cdot N}{(1-\sigma) + \sigma \cdot N \cdot (1+\lambda)}$. We have

$$\frac{\partial L(N, \lambda)}{\partial N} = \frac{\sigma[(1-\sigma) + \sigma \cdot N \cdot (1+\lambda)] - \sigma \cdot (1+\lambda) \cdot [(1-\sigma) + \sigma \cdot N]}{[(1-\sigma) + \sigma \cdot N \cdot (1+\lambda)]^2} = \frac{-\sigma \cdot \lambda \cdot (1-\sigma)}{[(1-\sigma) + \sigma \cdot N \cdot (1+\lambda)]^2} < 0.$$

$L$ in this case is a decreasing function of $N$, with a lower limit $L_l = \frac{1}{1+\lambda}$, which provides yet another evidence of the optimality
of $\lambda = 1$.

Experiment results show that in the case of ICON-O-HAMOCC $L$ is in general a decreasing function of $N$ (see Section 5),
and concurrency is effective only after a scaling threshold has been reached. Scaling tests for the ICON-A also indicate that
$L(N, \lambda)$ is a decreasing function of $N$, see Giorgetta et al. (2022), Table 3.

Let us now examine the concurrency communication cost $C$. The load of the point to point communication is proportional to
$1/N$, unlike to the halo communication cost, which is proportional to $1/\sqrt{N}$. So we do not expect the ratio $C(N \cdot (1+\lambda))/T(N)$
to change significantly as a function of $N$, but load imbalance, interconnect and latency costs may influence it. In some cases
concurrency may also require a halo exchange (this is the case for ICON-O-HAMOCC). Concurrency, obviously, will perform
better when the relative cost of the communication $C$ to the computational cost $T_A$ is small. If we take $C(N \cdot (1+\lambda))/T(N) = c$, constant then $\frac{T_c}{T_d} = L(N, \lambda) \cdot (1+c)$, which implies that the scaling threshold $L(N, \lambda) < \frac{1}{1+c}$ has to be reached before
concurrency is effective.

The case of super-linear scaling is rare but not unknown. Typically it occurs on cache-based architectures, when smaller
memory footprint allows more efficient use of the cache memory. We can still use Eq. 1 to deduce some conclusions (we can
switch modules $A$ and $B$ if necessary). In this regime we have $F(N_1) < F(N_2)$ for $N_1 < N_2$, and $L(N, \lambda) > 1$; so concurrency
will result in worse performance.

Finally we examine the case of "flattening" scaling, the limit after which the speed-up does not increase. This can be the
case for massively multicore architectures, like GPUs, when the workload per node is not enough to occupy the computing
units, and resources are idling. Let $N$ be the number of nodes beyond which scaling does not increase. Then, from the previous
analysis, we see that in the concurrent case scaling can still be increased up to $N \cdot (1+\lambda)$ nodes, but not beyond this. This in
essence underlines the fact that concurrency increases the parallel workload when compared to data parallelism.





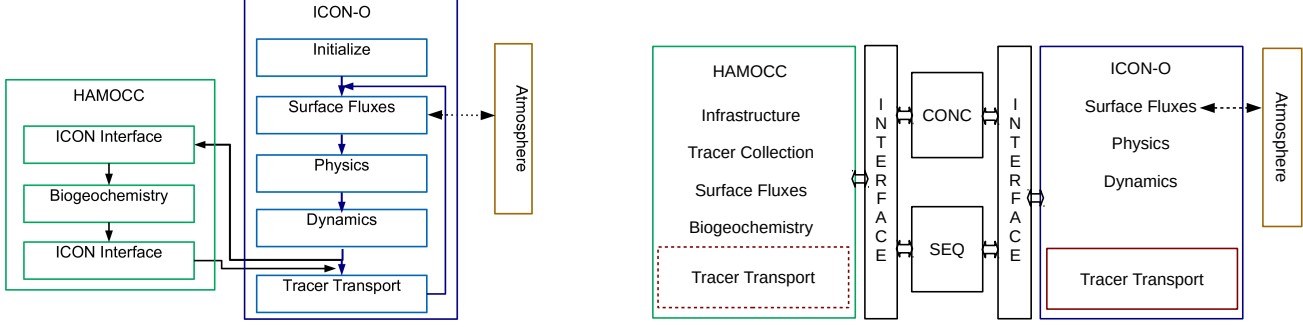

**Figure 5.** Left: diagram of the sequential ICON-O-HAMOCC flow. Right: diagram of the concurrent ICON-O-HAMOCC flow. The same interface is used for both the sequential and the concurrent mode.

## 4 Engineering concurrency for the ICON-O-HAMOCC model

Constructing coarse-grained component concurrency is a software engineering task. The candidate components have to present "natural" concurrency; this in practice means that they will present one communication point between them, while the algorithmic part of the components can run independently. Having identified the two components, the next steps are a. encapsulating the components, b. creating an interface between them, and c. providing the necessary infrastructure for the two components to run independently and to communicate. The procedure is not dissimilar to that of constructing stand-alone models, as described

for example in Eastham et al. (2018), or constructing coupled setups, as described in Long et al. (2015).

The original structure and workflow of the sequential ICON-O HAMOCC process is sketched in Fig. 5, left. The call to the HAMOCC biogeochemistry takes place just before the tracer transport is called (we will use the term transport for brevity). Upon returning, the HAMOCC tracers have been updated regarding the biogeochemistry processes, and the tracer transport routine is called. The HAMOCC tracers are transported along with the other two ICON-O tracers, temperature and salinity.

In this scheme ICON-O and HAMOCC are entangled through the memory usage and the tracer transport. The HAMOCC tracers are part of the ICON-O tracer structure. Other HAMOCC variables, like tendencies and sediment, while exclusive to HAMOCC, were still created in ICON-O in order to allow the use of the ICON infrastructure, like I/O. The ICON-O-HAMOCC interface handles the memory recasting between ICON-O and HAMOCC, as they use different memory layouts. The surface fluxes for HAMOCC are also handled in ICON-O. On the other hand, ICON-O does not have any dependencies on HAMOCC,

except optionally the calculation of solar short wave radiation absorption ratio, which is calculated in HAMOCC based on the chlorophylls concentration.

While the two components are entangled, the basic prerequisites for concurrency exist: algorithmic independence, and one point communication. There is though a point for further consideration: most of the time when running ICON-O-HAMOCC is actually spent in the tracer transport, rather than in the HAMOCC biogeochemistry itself (see Section 5). HAMOCC transports

17 tracers, making it the most expensive part in the ICON-O-HAMOCC execution. Parallelizing only the HAMOCC biogeo-



chemistry would result in only modest performance benefits, as the bulk of the execution would still be sequential (following the discussion in section 3.2).

The solution that we follow is to allow the biogeochemistry to transport its own tracers, independently of the ocean. This requires the encapsulation of the tracer transport, so that it can be called by both ICON-O and HAMOCC. Two structures were created as interface to the tracer transport: a. A tracer collection structure, that contains the information of the tracers required for the transport, and b. A transport state structure, that contains all the required fluxes. The transport state can be communicated from ICON-O to HAMOCC, allowing it to run the transport independently of ICON-O.

The tracer transport is an "embarrassingly" parallel process with regard to the number of tracers; every tracer can be transported independently of the others. This offers another level of data-parallel, medium-level parallelism, but it also presents some technical challenges, such as how to efficiently handle multiple communicators. We note that this approach does not replace concurrency, as large part of the code, notably the dynamical core, still runs sequential, presenting a scaling bottleneck. This level of parallelization has not been implemented in this project.

The next task was to disentangle the HAMOCC memory from ICON-O. The memory management of HAMOCC was moved into the HAMOCC component, and references to common global memory between the two components were removed. The surface flux calculations for the biogeochemistry were also moved from the ICON-O surface module to HAMOCC.

In the next step an interfacing mechanism between ICON-O and HAMOCC was created. This mechanism passes as parameters the information required for the two models. To HAMOCC the ocean and atmosphere variables are passed as described in Section 2.2; in addition the transport state is passed to HAMOCC to be used by the HAMOCC tracer transport. To ICON-O the short wave penetration is passed, and in the of case a coupled setup, the $CO_2$ fluxes, which in turn are passed to the atmosphere through the coupler.

The final task was to provide HAMOCC with the necessary infrastructure to run autonomously. ICON provides a simple mechanism through namelists for registering the components that run concurrently, by defining the component, and the group of MPI processes assigned to it. The calls to the infrastructure setup, such as domain decomposition, setting the communicators, the I/O, and the coupler, is done in the initialization phase in each of the components. While this mechanism does not provide the sophistication and power of more complex infrastructure frameworks, like the Earth System Modeling Framework (Hill et al. (2004); Collins et al. (2005)), and it requires to partly duplicate the code of setting-up the infrastructure, it provides high level infrastructure interfaces and it is serviceable.

The final construction is presented in Fig. 5 right. Two interfaces are constructed to send and receive information between ICON-O and HAMOCC on each side. The communication takes place just before the tracer transport for ICON-O, while for HAMOCC at the beginning of each timestep. The information communicated form ICON-O to HAMOCC includes the temperature, salinity and pressure, used in the chemistry processes, surface fluxes, such as the total surface water flux, the solar radiation flux, the $CO_2$ concentration, and the wind stress. For the tracer transport, HAMOCC receives the fluxes and velocities from ICON-O, as well as the sea surface height. On the other side, ICON-O receives optionally the solar radiation absorption ratio, and in the case of a coupled setup with the atmosphere the $CO_2$ fluxes, which in turn are sent to ICON-A.





The interfaces can serve two modes: sequential, or concurrent. Both modes are transparent, the two components are "un-aware" of the mode they run, as this is handled within the interfacing mechanism. This process also works in the coupled ICON-O-HAMOCC ICON-A setup, so the three components can run concurrently.

The interfacing mechanism serves only to communicate information between the two components on the same grid, without providing any further functionality that general couplers may provide. In the concurrent mode we use the communication li-
brary YAXT (Yet Another eXchange Tool, https://swprojects.dkrz.de/redmine/projects/yaxt) developed at DKRZ. It provides a flexible interface that allows to define both 2D and 3D communication patterns, and can also aggregate the communication into one call. YAXT provides an abstraction for defining communication without any explicit MPI-message-passing calls. The communication scheme is automatically derived from descriptions of locally available data on each process. Thus, very different communication patterns, like transpositions or boundary exchanges, can be generated in a user friendly manner, independently
of the complexity of the domain decomposition. This leads to a significantly reduced and less error-prone programming effort. YAXT can also be used to generate the communication patterns for the redistribution of data between two sets of processes that use different domain decompositions, but the same grid. This is an essential functionality for implementing concurrency communication between components, that do not necessarily share the same decomposition, but they share the same grid.

The new ICON-O-HAMOCC implementation gives bit identical results with the original one, both in the sequential and
concurrent mode. It has also been technically checked for correctness when running with an interactive atmosphere ocean carbon cycle, but the impact of concurrency in this setup still needs to be evaluated.

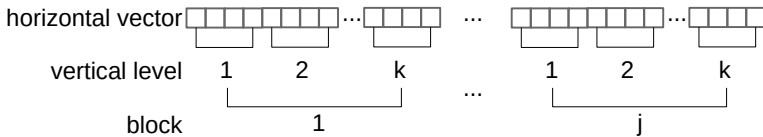

**Figure 6.** Schematic of the memory layout of ICON. In this example the vector size is four, but varies via a namelist parameter

A final step was taken to introduce OpenMP directives in HAMOCC. While this is not directly related to coarse grained concurrency, it provides the shared memory level of parallelization. The ICON memory layout consists of an inner vector chunk that corresponds to the horizontal dimension of the grid. Each vector chunk is assigned to a single vertical level, and a
set of full vertical levels and vectors constitute a block (see Fig 6). Array indexing takes the form $(i, l, b)$, with $i$ iterating the vector part, $l$ the vertical levels, and $b$ the blocks. The ICON memory consists of multiple blocks, and OpenMP parallelization is applied over these blocks. On the other hand, the HAMOCC memory layout consists of a single block. The whole HAMOCC code is executed inside a loop over blocks, copying in and out the respective ICON block. OpenMP parallelization is applied on this loop. We note though that this memory layout was designed primarily for vector optimization, and it is sub-optimal for
cache-based machines. It offers poor data locality for stencil operators, which impacts negatively the OpenMP performance and scaling.

A first study on the impact of the memory layout on performance and direct vs indirect (unstructured grid) indexing is presented in MacDonald et al. (2011), but without a discussion on its impact on shared memory parallelization and scaling.





## 5   Experiments and performance results

We have performed two sets of experiments, one at a low horizontal resolution of 160km on a 36 cores-per-node machine, and another at a medium resolution of 40km on a 128 cores-per-node machine. The two setups were measured for strong scaling, both in sequential and concurrent mode. Each of the runs was repeated three times, and the best of the three, in terms of the total time, was selected for the analysis.

We study the behavior of the combined MPI, OpenMP parallelization and the coarse-grained concurrency. Most of the ICON-
O code is OpenMP parallelized, but not all. In particular, the sea-ice dynamics is not OpenMP parallelized, and it was disabled in order not to distort the scaling behavior. The Gent-McWilliams and the Redi parameterizations have not yet been included in the concurrent version, and they were also disabled. As we focus on the scaling behavior, with and without concurrency, rather than the performance itself, these two modules would not change our conclusions on the effect of the concurrency[4]. All output was disabled in these runs. The time measures do not include the initialization phase of the models, as this is a one-time cost,
and would distort the scaling analysis for short runs.

In fine-tuning a setup for performance we would typically calculate the number of OpenMP threads, the vector size, and the MPI tasks, so that shared memory parallelization is balanced. No such effort was taken in these setups. We did examine though the effect of different vector sizes on the low resolution experiment.

A discussion on scaling bottlenecks for a similar ocean-biogeochemistry setup is presented in Epicoco et al. (2016).

### 5.1   Low resolution experiment, 160km

The 160km grid consists of 14298 horizontal ocean cells, and 40 vertical levels. This setup was run on the Mistral compute2 partition at DKRZ. Each node is equipped with two Intel Broadwell cpus, providing a total of 36 cores. The experiments ran for five simulated years.

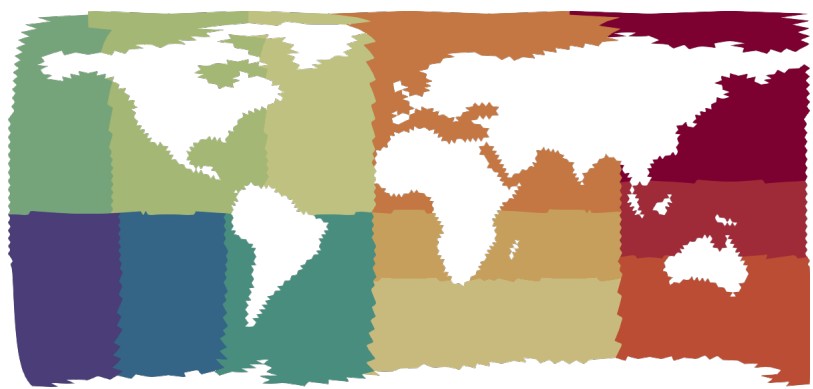

**Figure 7.** The 160km grid decomposed into 12 subdomains. .

---

[4] In fact, we expect that concurrency would be improved by including these two modules, as $\lambda$ would get closer to 1 (see also Table 1).



For the domain decomposition a recursive weighted medial decomposition is employed. Each subdomain is assigned the
number of total further "cuts", and is bisected in a weighted manner across the longest axis of weighted longitudes or latitudes.
For example, if a subdomain is to be decomposed into 5 subdomains, the longest axis is found, and is bisected with weights 2
and 3. The two child subdomains are assigned 2 and 3 cuts respectively, and the process continues recursively. An example of
the domain decomposition is given at Fig. 7.

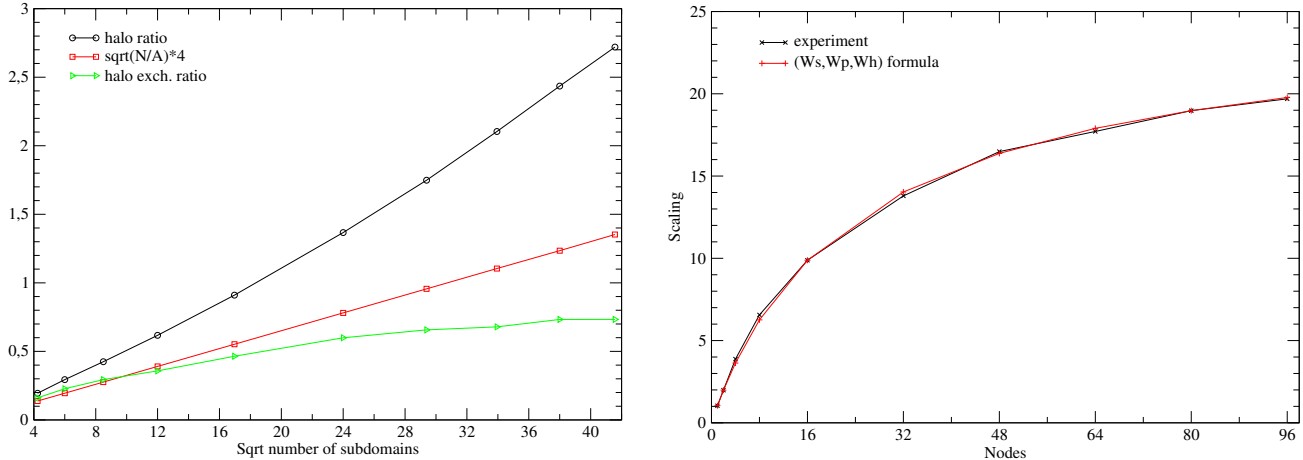

**Figure 8.** Left: the ratio of halo to non-halo cells. Right: scaling results from the 160km experiment, and scaling computed from formula 2.

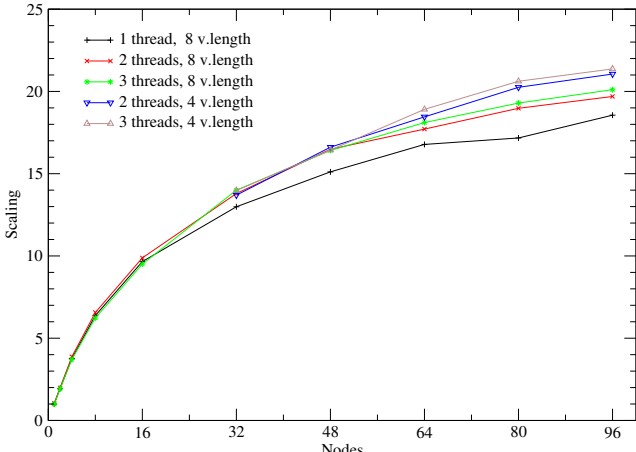

**Figure 9.** Scaling for the sequential 160km setups for different numbers of OpenMP threads and vector lengths.

First, we examine the behavior of the sequential experiments. In the case that the domain decomposition would produce
perfectly balanced square subdomains, the number of halo cells per subdomain would be proportional to $4\sqrt{A/N}$, where $A$
is the total number of grid cells and $N$ the number of subdomains. The total number of halo cells would be proportional to



$4\sqrt{N \cdot A}$. In Fig. 8 left, the ratio of halo cells to the grid cells is depicted, as well as the ratio of $4\sqrt{N \cdot A}$ to $A$. The actual halo ratio is significantly larger than the ideally calculated one, due to the imperfect decomposition with regard to the number of halo points. The relative cost of the halo communication is also depicted, which only partly reflects the increase of halos.

In ICON, whenever possible, halo values are computed, instead of communicated, unless the computation is expensive. So the total computation cost is increased as well.

We can further approximate the scaling behavior of the experiment by considering three types of workload: a sequential part $W_s$, a purely parallel part $W_p$ and a halo part $W_h$. We have for the total workload $W_t = W_s + W_p + W_h$, where we measure it in terms of time cost. The time cost when running on $N$ MPI processes (i.e. subdomains) is

$$T_N = W_s + W_p/N + W_h/\sqrt{N}. \tag{2}$$

We estimated the workloads based on the sequential runs on 2, 16 and 80 nodes as: $W_s = 243$ secs, $W_p = 101535$ secs and $W_h = 774$ secs. In Fig. 8 right, the scaling results from the experiments and the above formula are shown. The formula captures well the scaling behavior of the experiments, highlighting the importance of the non-scaling parts of the code.

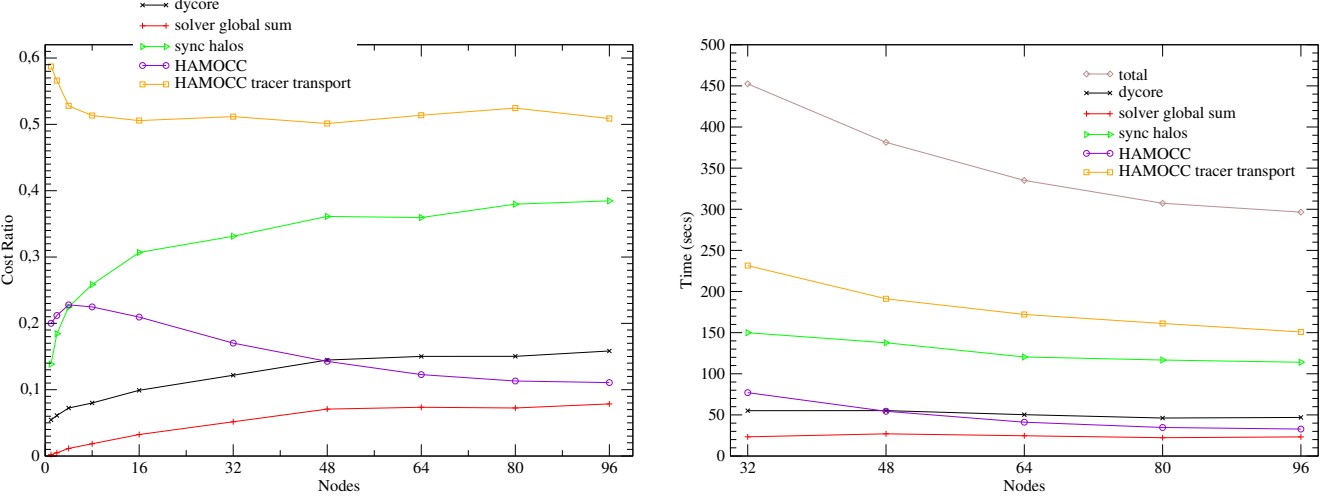

**Figure 10.** Left: relative cost ratio of the ICON-O-HAMOCC components in the sequential 160km setup. Right: run time of the components.

For this experiment we tested the scaling for 1, 2, 3, 6 and 9 OpenMP threads. More than 3 threads performed worse, and we

do not include them in the results. We also checked the impact of a vector length 8 and 4 to the scaling behavior. The results are presented in Fig. 9. The OpenMP parallelization becomes more important for higher number of nodes. The vector length of 4 provides finer granularity and better balancing in higher number of nodes, and so it improves performance. We note that even as the number of halo cells exceeds the number of the original grid cells, we still get some scaling. As discussed above, the theoretical scaling in the case of the halo computation being dominant would still be proportional to $\sqrt{N}$.

Next we will only consider the best of the runs from Fig. 9 (that si the ones with the smallest total time). The relative costs and the times for the major components of the model are presented in Fig. 10. The highest cost comes from the HAMOCC

tracer transport, while HAMOCC itself incurs relatively small cost. The halo communication cost imposes the most important restriction to scaling. This also includes load imbalancing costs, which increase with the number of subdomains. The dycore (dynamical core) process consists of the calculation of the horizontal velocities and the sea surface height. ICON-O uses the iterative conjugate gradient method (CG) for inverting the matrix required by the implicit sea surface height calculation. In each iteration the computation of a global sum is required for calculating the magnitude of the residual. The time cost for the global sum used by the CG solver remains constant[5], imposing a sequential scaling limit, as described by Amdahl's law. Its relative cost though remains small in these runs due to the large cost of the tracer transport.

**Figure 11.** Scaling (solid line) and parallel efficiency (dashed line) for the 160km sequential and concurrent setups. The 0.5 efficiency line is marked.

**Figure 12.** Left: relative costs of the ICON-O components in the concurrent 160km setup. Right: run times of the ICON-O components.

[5] This cost is included in the dycore cost.

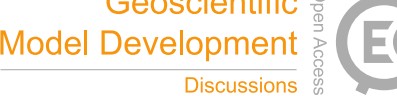

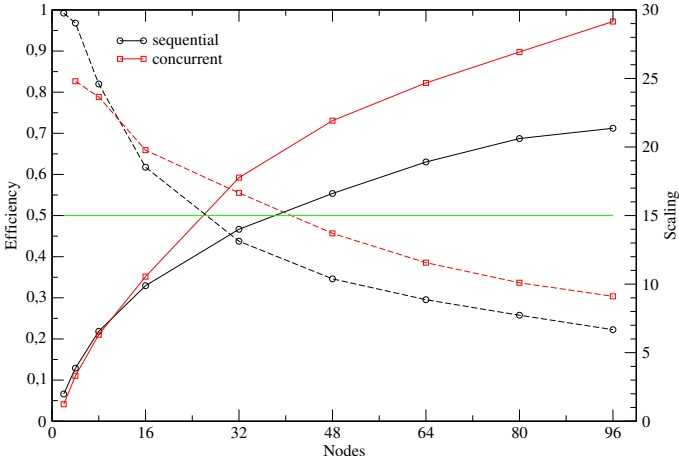

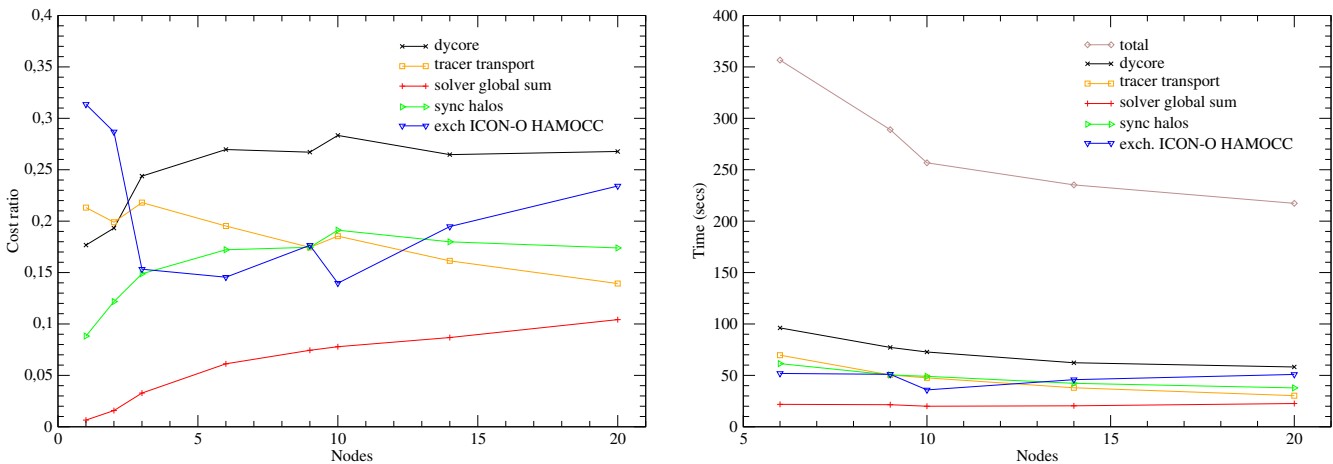





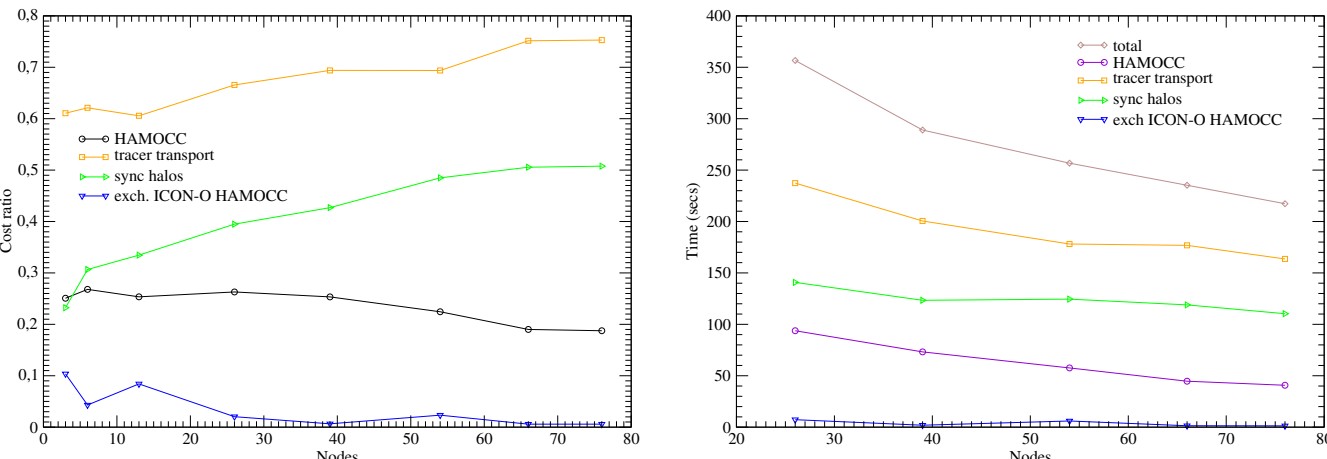

**Figure 13.** Left: relative costs of the HAMOCC components in the concurrent 160km setup. Right: run times of the HAMOCC components.

In the concurrent setup we kept the number of OpenMP threads constant, equal to 2. The vector length was set to 8, except
for the experiments on 80 and 96 nodes, where it was set to 4. The MPI tasks are organized into two contiguous groups, the first containing the ICON-O tasks and the second the HAMOCC tasks. The scaling behavior of both the concurrent and the sequential case is depicted in Fig. 11. The performance is improved only after the sequential parallel efficiency has gone down to about 60%. The limit of 50% efficiency is reached in the sequential case at about 24 nodes, while in the concurrent at about 40 nodes.

In Fig. 12 and 13 the relative costs and timers for the components of ICON-O and HAMOCC respectively are given for the concurrent experiments.

The different scaling characteristics of the two modules create additional imbalance, some care has been taken to improve this by reducing the number of ICON-O nodes, see Table 1. The imbalance becomes apparent in the cost of exchanging data and synchronizing ICON-O and HAMOCC. This cost in ICON-O is large, reaching more than 20%, while in HAMOCC is
negligible. This indicates that the actual communication cost is small, while the cost of imbalance is beared by ICON-O. This imbalance though does not affect significantly the total performance, as on a total of 96 nodes for example, 76 are used for HAMOCC, and increasing this number by a few nodes would result little improvement.

In section 3.2 an analysis was provided regarding the scaling characteristics of concurrency in relation to the sequential parallel efficiency. We expect that concurrency would be beneficial after a parallel efficiency threshold has been reached, and
this is confirmed by the experimental results. In Table 1 we compare the values of $T_c/T_d$, as computed by formula 1 in Section 3.2, and the ones given by the experiments. In Section 3.2 we defined the workload as the total number of operations, and $\lambda$ as the ratio of the workload between component $A$ and $B$; both of them were considered to be constant numbers, independent of the number of nodes. We further assumed that the scaling behavior of $A$ and $B$ is the same. In our experiment setup though the situation is different. The scaling behavior of the two models is different, with the ICON-O scaling being worse that HAMOCC.
In the sequential runs, the total relative cost of HAMOCC (including the HAMOCC tracer transport) is about 80% on 2 nodes,





| Total Nodes | ICON-O Nodes | $\lambda_{exp}$ | $T_c/T_d$ , $\lambda = 3$ | $T_c/T_d$, $\lambda = 2$ | $T_c/T_d$, $\lambda = 1$ | $T_c/T_d$, $\lambda = \lambda_{exp}$ | $T_c/T_d$ exp. |
|---|---|---|---|---|---|---|---|
| 4 | 1 | 3.10 | 1.07 | 1.11 | 1.07 | 1.07 | 1.17 |
| 8 | 2 | 2.82 | 1.01 | 1.01 | 0.93 | 1.01 | 1.04 |
| 16 | 3 | 2.51 | 0.97 | 0.94 | 0.83 | 0.96 | 0.94 |
| 32 | 6 | 2.14 | 0.94 | 0.91 | 0.78 | 0.92 | 0.79 |
| 48 | 9 | 1.81 | 0.92 | 0.90 | 0.74 | 0.86 | 0.76 |
| 64 | 10 | 1.75 | 0.94 | 0.87 | 0.74 | 0.84 | 0.77 |
| 80 | 14 | 1.77 | 0.98 | 0.83 | 0.74 | 0.81 | 0.77 |
| 96 | 20 | 1.63 | 0.95 | 0.82 | 0.71 | 0.78 | 0.73 |

**Table 1.** The first two columns describe the nodes used for the ICON-O-HAMOCC concurrent setup. Columns 4–6 describe the estimated ratio of concurrent to the sequential time $T_c/T_d$ calculated using Equation 1, and the relative efficiency for different values of $\lambda$ from the sequential runs. In column 3 $\lambda = \lambda_{exp}$ is computed from the sequential runs, and in column 7 the estimation of the $T_c/T_d$ ratio for this $\lambda$ through interpolation. In the last column the actual $T_c/T_d$ is computed from the runs.

while on 96 nodes this cost is about 62%. The relative cost of ICON-O almost doubles in this range. In order to account for this difference, we will consider $\lambda$ not to be constant, but dependent on the number of nodes, so that it reflects the relative cost of HAMOCC in the sequential experiments. We set $\frac{C(N \cdot (1+\lambda))}{T(N)} = 0.1$ as a first guess. We calculate $L(N, \lambda) = \frac{F(N \cdot (1+\lambda))}{F(N \cdot \lambda)}$ from the sequential experiments for $\lambda = 1, 2, 3$, and the corresponding ratio $T_c/T_d$ is given in columns 6,5,4 respectively. We note

that $\lambda \geq 1$, so we have $L(N, \lambda) = \frac{F(N \cdot (1+\lambda))}{F(N \cdot \lambda)}$. In column three we give an estimation of $\lambda_{exp}$ based on the sequential runs, computed as the ratio of the HAMOCC time to the ICON-O time.

In column seven the estimation of the concurrent to the sequential time ratio $T_c/T_d$ is given by linearly interpolating the $T_c/T_d$ values from the $\lambda = 1, 2, 3$ parameter to the $\lambda_{exp}$. In the last column the actual ratio $T_c/T_d$ is given from the experiments. While formula 1 is a great simplification of the real model behavior, it still gives a reasonable approximation of the scaling

behavior of the concurrent setup, based on the behavior of the sequential one. Furthermore, by comparing columns 4, 5 and 6, we observe that the predicted concurrency efficiency declines when $\lambda$ deviates from the optimal value of one, as expected, and the experiments $T_c/T_d$ similarly improves as $\lambda_{exp}$ approaches 1.

## 5.2 Medium resolution experiment, 40km

The 40km grid consists of 230124 horizontal ocean cells, and 64 vertical levels. The experiments ran for one simulated year.

They were run on the Levante machine at DKRZ, each node is equipped with two AMD 7763 CPUs, giving a total of 128 cores per node. The number of OpenMP threads is constant, equal to 4, the vector length is equal to 8. The MPI tasks are placed in a cyclic way in groups of eight across the nodes, so that the first eight MPI tasks occupy half of the first CPU of the first node, the second group occupies half of the first CPU of the second node, and so on. In the concurrent case the group of eight tasks consists of two ICON-O tasks and six HAMOCC tasks. This placement alleviates load imbalancing, as it allows improved

memory bandwidth for the slower processes.





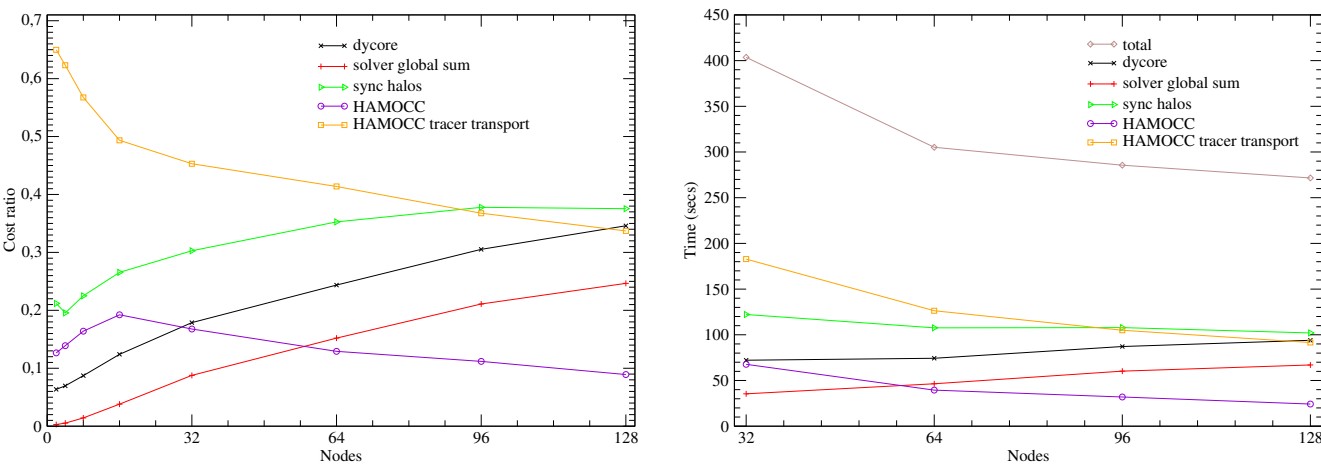

**Figure 14.** Left: relative costs of the ICON-O-HAMOCC components in the sequential 40km setup. Right: run times for each component.

In Fig. 14 the sequential setup time costs are presented. The picture differs from the 160km setup in the time cost of the CG global sum, which here takes 25% of the time on 128 nodes. The total communication cost, including the global sum and the halo exchange, on 128 nodes exceeds 60% of the total time, while for the 160km on 96 nodes this cost is about 47%. This underlines the communication bottlenecks when using machines equipped with powerful multicore nodes.

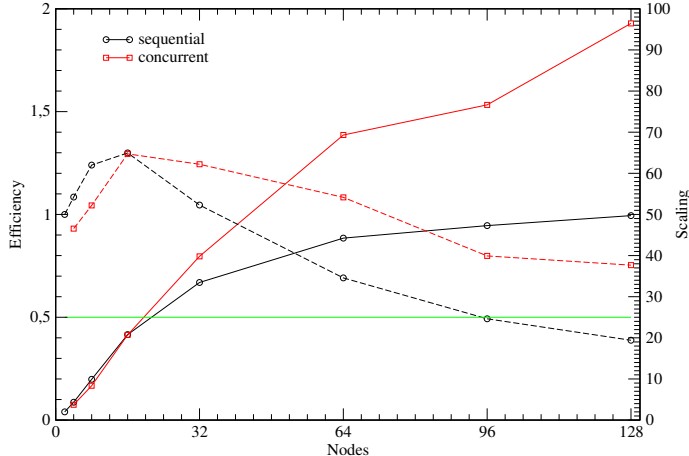

**Figure 15.** Scaling (solid line) and parallel efficiency (dashed line) for the 40km sequential and concurrent setups. The 0.5 efficiency line is marked.

The scaling of the sequential and the concurrent setup is shown in Fig. 15. Scaling is measured against the sequential run on two nodes. The first thing to observe is the super-linear scaling of both the sequential and the concurrent setups when using up to 16 nodes. This behavior typically is caused by more efficient cache usage in NUMA machines. An indication towards this inference is provided from the behavior of the HAMOCC tracer transport in Fig. 14 left. The tracer transport process is memory



intensive due to the large number of stencil operations, which, as we have noted, are sub-optimal due to poor data locality. Its
cost drops sharply when using up to 16 nodes, which indicates better relative memory efficiency. Another related consequence
is that the parallel efficiency in the sequential run reaches the 0.5 mark at a relatively high count of nodes, 96, while in the
concurrent case it never reaches this limits, and stays above 0.7. This is due to the poor performance on the reference number
of nodes of two.

We note that formula 2 does not apply to this experiment. This is partly due to the superlinear behavior of this setup in the
low count of nodes, but also because this setup exhibits a part that its cost increases with the numbers of nodes, namely the
global sum (see Fig. 14 right). This cost is significant, and is not been accounted in formula 2.

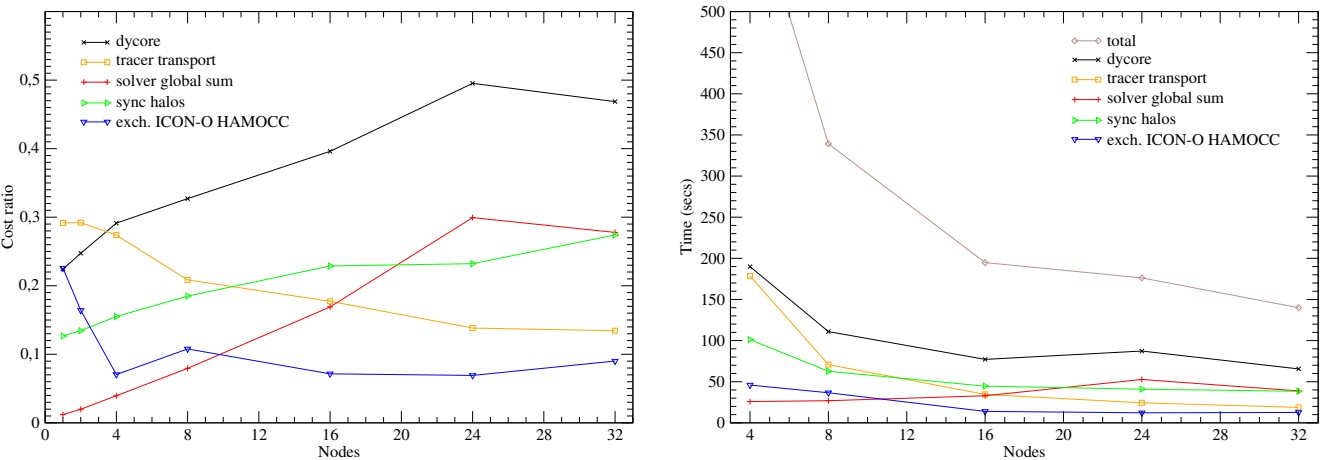

**Figure 16.** Left: relative costs of the ICON-O components in the concurrent 40km setup. Right: run times for each component.

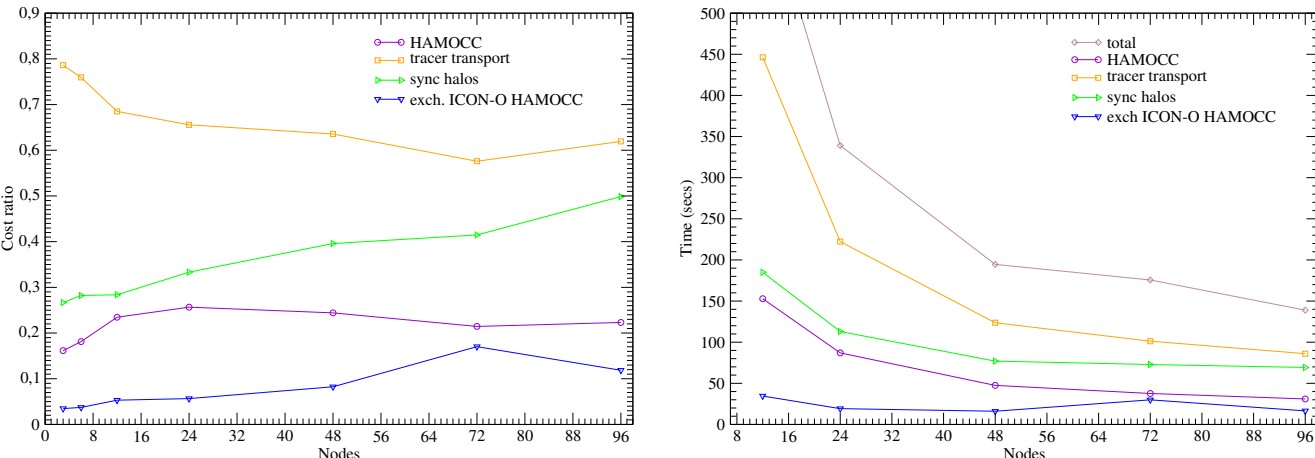

**Figure 17.** Left: relative costs for the HAMOCC components in the concurrent 40km setup. Right: run times for each component.





| Total Nodes | $\lambda_{exp}$ | $T_c/T_d$, $\lambda=3$ | $T_c/T_d$, $\lambda=2$ | $T_c/T_d$, $\lambda=1$ | $T_c/T_d$, $\lambda=\lambda_{exp}$ | $T_c/T_d$ exp. |
|---|---|---|---|---|---|---|
| 4 | 3.20 | 1.17 | 1.12 | 1.19 | 1.17 | 1.17 |
| 8 | 2.73 | 1.21 | 1.14 | 1.26 | 1.19 | 1.19 |
| 16 | 2.18 | 1.11 | 1.14 | 1.15 | 1.14 | 1.00 |
| 32 | 1.64 | 0.97 | 1.01 | 0.89 | 0.96 | 0.84 |
| 64 | 1.19 | 0.90 | 0.89 | 0.73 | 0.76 | 0.64 |
| 96 | 0.92 | 0.81 | 0.87 | 0.64 | 0.64 | 0.62 |
| 128 | 0.74 | 0.87 | 0.78 | 0.61 | 0.61 | 0.52 |

**Table 2.** As Table 1, for the 40km experiment. The ratio of the ICON-O to the HAMOCC cores is constant, 3/1, and is omitted.

The relative costs and times for ICON-O and HAMOCC in the concurrent setup are given in Fig. 16 and 17 respectively. We again see that the communication costs within each module is significant, while the coupling cost between ICON-O and HAMOCC remains relatively small.

In Table 2 we perform the same calculations for the 40km setup, as we did in Table 1 for the 160km setup. We see a similar picture, except for the first two rows, where both the predicted and the actual concurrency efficiency declines as a function of $N$ and as a function of $\lambda$. This a result of the super-linear scaling. Only after the threshold of 16 nodes concurrency becomes beneficial, and thereafter its efficiency, relatively to the sequential setup, increases. While the theoretical prediction underestimates the effect of concurrency (this is due to the different scaling behavior of the two components, which is not 485   accounted in the formula), it still gives a reasonable estimation.

    We note that the concurrency efficiency is significantly better in this setup, compared to the 160km. The ratio $T_c/T_d$ reaches 0.52, while for the 160km it is 0.73. This behavior reflects the limitations of the data parallel approaches when using highly parallel architectures. Fig. 14 reveals the issue: the cost of communicating the halos dominates the performance at 128 nodes, along with the solver global sum.

| | | 160km, 1152 Mistral cores | 40km, 16384 Levante cores | Ratio 40km/160km |
|---|---|---|---|---|
| 1 | 2D cells | 14298 | 230124 | 16.10 |
| 2 | 3D cells | 481402 | 11960498 | 24,85 |
| 3 | 2D cells/core | 12.41 | 14.05 | 1.13 |
| 4 | 3D cells/core | 417.88 | 730.01 | 1.75 |
| 5 | Timestep (min) | 60 | 45 | 0.75 |
| 6 | 3D cells/core/sim.min. | 6.97 | 16.22 | 2.33 |
| 7 | Seq. time (sec), 1 SY | 90.50 | 271.63 | 3.00 |
| 8 | Conc. time (sec), 1 SY | 71.22 | 140.00 | 1.97 |

**Table 3.** Detailed numbers for the two experiments selected from the two setups of 160km and 40km. The description is provided in the text. 1152 Mistral cores correspond to 32 nodes, while 16384 Levante cores correspond to 128 nodes.





The two setups of the 160km and the 40km present a scenario where the resolution is increased, in combination with the transition to a more powerful and parallel machine. We examine the impact of concurrency in this scenario. Two experiments were selected from the two setups, so that they have approximately the same number of 2D cells per core, and at the same time they are at similar scaling limit in the sequential case of each setup, where the parallel efficiency drops below 50%. The details of these two experiments are given in Table 3.

In row 6 the number of 3D cells per core, normalized for one simulated minute, is given. Their ratio suggests the relative workload per core for the two experiments: the 40km has 2.3 times more load per core[6]. In rows 7 and 8 the times for simulating one year are given. In the sequential case, the 40km is 3 times slower than the 160km, a value that is above the estimated 2.3. On the other hand, in the concurrent case, the 40km experiment is less than 2 times slower, which provides another indication of the increased impact of concurrency when moving to highly parallel machines.

## 6   Discussion and outlook

By decomposing the algorithmic space, coarse-grained component concurrency offers another parallelization dimension, in addition to the existing data parallel approaches. It improves scalability when certain scaling limits have been reached through the data parallel methods. It is more effective in the regimes where the relative parallel efficiency drops. In the regimes of linear scaling it results in little, if any, improvements. It produces higher parallel workload when compared to data parallel approaches. This makes it suitable, and probably indispensable, for efficiently utilizing massively parallel machines. Our experiments show that the concurrency effectiveness increased from 1.4 times improvement on a 36 cores-per-node machine to 2 times improvement on a 128 cores-per-node machine. We also see that the "coarseness" is an important factor to this approach. It gives the best results when the two components incur approximately the same cost, while its effectiveness deteriorates when moving away from this balance. Both our theoretical analysis and our experimental results concur to the above conclusions.

It is clear that coarse-grained concurrency does not make the code faster, and the more traditional code optimization and scaling improvement procedures are still an important part of the process of getting the models to run efficiently. We expect for example that improving data locality for ICON-O-HAMOCC would have a higher performance impact on Levante than concurrency. These more intricate optimization processes though require a high level of expertise and in cases extensive code restructuring. In comparison, engineering coarse-grained component concurrency is in general a simpler process, and requires only a modest effort when good software engineering practices were already in place.

Coarse-grained concurrency can always be applied on top of the other optimizations, allowing us to use more, and more efficiently, computational resources. This implies that enough parallel computational resources should be available in order to be practically useful. On the other hand, the new machines are massively parallel, with nodes equipped with hundreds or thousands of cores, providing computing power that has reached the exaflop level. The most effective way to make use of this massively parallel computing power is through multi-level and multi-dimensional parallelism. Highly parallel architectures, like GPUs, require a minimum of parallel workload to provide the best efficiency. Experiments show that we need millions of

---

[6]Some details are not taken into account here, such as the number of iterations the CG solver requires, which is higher for the 40km.

3D grid points per GPU to make full use of them (Leutwyler et al. (2016); Giorgetta et al. (2022)). This limits the extend that data parallelism can be used on such machines. We expect that coarse-grained concurrency will provide an effective leverage for making use of these architectures.

Coarse-grained concurrency is a high level parallelization, and thus independent of architectures. This makes it applicable on different architectures, without the need to re-implement the concurrency mechanism. Furthermore, it can be applied on heterogeneous environments in hybrid mode, as on CPUs and GPUs, providing the potential to make use of all the resources of heterogeneous machines.

    A positive side-effect of coarse-grained component concurrency is that it naturally bequeaths concurrency to the infrastruc-
ture attached to these components, like I/O and real-time postprocessing. Especially I/O can pose a significant performance bottleneck, and parallel asynchronous I/O approaches have already been developed, see for example Brown et al. (2020); Yepes-Arbós et al. (2022); Hohenegger et al. (2022). The naturally inherited concurrency to the components' infrastructure can further enhance the performance of such schemes. While these side-effects have not been the subject of this paper, we expect them to help increase the efficiency of the existing approaches.

The applicability of coarse-grained concurrency seems plausible to other components of Earth system models. Its effective-ness will depend on the components' workload, how much "coarseness" they present, and how tightly they are connected, which will reflect on the coupling cost. In some cases, questions related to the handling of the feedbacks between the com-ponents have to be considered. We have not addressed this question regarding the concurrent ICON-O-HAMOCC when the interactive carbon cycle is activated in a coupled setup. This would be a subject of future work.

*Code availability.* The ICON code is available under licenses, see https://mpimet.mpg.de/en/science/modeling-with-icon/code-availability. The experiments were performed with ICON v.2.6.5. The code is also documented in the provided dataset.

*Data availability.* The code, scripts and results are available in the dataset provided in Linardakis (2022).

*Author contributions.* I. Stemmler integrated the HAMOCC code into the ICON framework and contributed to the concurrent implemen-tation. M. Hanke provided the YAXT library and its interface for the concurrent ICON-O-HAMOCC communication, and contributed the
YAXT description. L. Ramme, F. Chegini and T. Ilyina contributed the biogeochemistry and HAMMOCC description. P. Korn provided the ICON-O description. L. Linardakis contributed the rest. All co-authors contributed to the text formulation.

*Competing interests.* The authors declare that they have no conflict of interest.



*Acknowledgements.* Marco Giorgetta offered many insightful comments that helped clarify the concepts presented in this paper. Daniel Klocke provided extensive suggestions for improving the paper. Reiner Schnur read a first draft of this paper and offered points for improve-

ment. Kai Logemann and Moritz Mathis provided the ICON-O-HAMOCC coastal ocean grid for illustration purposes. Jan Frederik Engels and Panos Adamidis at DKRZ have helped us with the technical aspects of running the experiments on Mistral and Levante.

T. Ilyina and F. Chegini were funded by the European Union's Horizon 2020 research and innovation program under grant agreement number 773421—project 'Nunataryuk' and by the Deutsche Forschungsgemeinschaft (DFG, German Research Foundation) under Germany's Excellence Strategy— EXC 2037 "Climate Climatic Change and Society" (CLICCS)—Project Number: 390683824.

T. Ilyina was supported by the European Union's Horizon 2020 research and innovation program under grant agreement No 101003536 (ESM2025—Earth System Models for the Future) and the German paleoclimate modelling initiative PalMod (FKZ: 01LP1505A, 01LP1515C). PalMod is funded by the Bundesministerium für Bildung und Forschung (BMBF), and it is part of the Research for Sustainable Development initiative (FONA).





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
