# Peer review of "Improving scalability of Earth System Models through coarse-grained component concurrency - a case study with the ICON v2.6.5 modeling system"

_Geoscientific Model Development, 2022_

## Author Comment (AC1)

Thank you for reviewing the paper and for your comments. I have adjusted the revised version of the paper accordingly, and here I provide the answers (in black) to your comments (in blue).

This is a good paper addressing the important topic of concurrency that has so far not been explored enough in Earth system modelling. The paper is well written but could be more concise. The theoretical analysis is backed up with convincing performance measurements that show the usefulness of concurrency. The paper will be ready for publication in GMD if the following minor comments are addressed.

More significant points:

- Page 9 seems to be a broken in the published PDF. I opened the document with 3 different PDF viewers, but I never got to see the entire page.

  I suspect that this relates to the pdf format of the images included in this page. I switched them to png format, and I will further ask GMD for advise.

- Overall, the document is longer than it would need to be. E.g. the model descriptions could be shortened as documentations to the models are provided as references. This is a subjective opinion by me, so feel free to disagree, but maybe you can consider making the document shorter.

  I also feel that the document overall is a bit extensive and I removed some details that seemed of small contribution:
    - Some details from the ICON-O model description were removed.
    - The detailed memory discussion, lines 353-361, has been shorten, and Fig. 6 has been removed.
  I maintained though a short description of the models, for the readers that wish to have a quick overview, without having to go through the literature.

- Section 3.2: This is a very interesting discussion. However, I am not sure how much of this is new and how much of this is common knowledge in computer science. If it is new, it should be made more prominent (e.g. in abstract and intro). If it is not new, maybe provide references more prominently.

  I am not aware of any similar discussion in the computer science literature. I added one sentence hinting this analysis in the abstract: "We study the characteristics of component concurrency and analyse its behaviour in a general context. The analysis shows that component concurrency increases the ``parallel workload", improving scalability under certain conditions.".

- Figure 5: I am not a big fan of the figure. The left side is supposed to show a serial workflow but is showing two boxes in "parallel". Why not add a time dimension to the figure and

streamline the left side into a single line of tasks? Maybe you can also use the nomenclature of Section 3.2 (W,N…)?

I improved somewhat the left (sequential workflow) figure. Now the arrows indicate more clearly the sequential execution workflow (which also shows the top-down time direction). The right (concurrent workflow) figure is more tricky to improve, but I think it is still useful to illustrate the new code structure, and the double interfacing mechanism.

- I am missing a bit more of a discussion what happens when the components are not strictly independent. E.g. when running the radiation scheme concurrently to the rest of the atmosphere model. The independence is quite a significant limitation for couple Earth system model components.

The components are not "strictly independent", in the sense that no dependencies exis between them. In general, I am not aware of any component in the ESM that is strictly independent, this would mean that it has no interaction whatsoever with any other component. In the case of HAMOCC, when it has only a one-way dependency on the ICON-O, the results are binary identical. When feedbacks are activated (ocean solar absorption and interactive $CO_2$), the results are not any more binary identical and the impact has to be evaluated. I have restructured the paragraph at line 349 as follows:

"The new ICON-O-HAMOCC implementation gives bit identical results with the original one, both in the sequential and concurrent mode, when the HAMOCC feedabacks (the ocean solar radiation absorption ratio and the ocean-atmosphere $CO_2$ fluxes) are disabled. Bit identical results cannot be obtained in the case these feedback are activated in the concurrent mode, due the the different workflow from the sequential mode. It has been technically checked though for correctness when running with these feedbacks activated. The impact of concurrency on the results in this case still needs to be evaluated."

This is no different from the concurrent radiation, in both cases the impact of concurrency has to be evaluated. The nature and magnitude of this impact may differ, but the underline procedure is the same. This paper though is too long to add such an important discussion, and we differ it to future work (the last paragraph in the Outlook Section refers to this).

- Paragraph starting with L369: These seem to be quite significant limitations of the approach. Could you also provide timing results with GM parametrisation, IO etc. to give the reader an impression how significant the limitations are?

Adding the GMRedi parameterization would create more balanced components, and we expect this would improve the performance of concurrency (we note so in the related footnote). The reason that we have not included the GMRedi in the concurrency is purely due to high software engineering complexity, and the large amount of effort to include it in the concurrent structure. In ICON we target non-parameterized (as far possible) setups, and especially eddy-permitting with high vertical resolution, so it is not at all clear if an

investment on the GMRedi parameterizations is worth it. We will examine it in the future, if deemed necessary.

The impact of IO on performance and scaling highly depends on the output volume and frequency, and can be very different for different output demands. This makes it a poor objective criterion for model scaling tests. On the other hand, in practice it is clearly a factor to take into account. We expect that concurrency will have a positive impact on IO, as it provides natural parallelism for the whole infrastructure, we discuss this in the paragraph L. 529. This approach is MPI based, in a similar way that we run the coupled atmosphere and ocean models, so all infrastructure calls, including output, are automatically component-parallelized too. I hope we will have some results on this in the future, but I feel this paper is already too extensive to include here further experiments.

- Can you say anything about the energy cost when using concurrency (or not)? No big problem if not.

As a general comment, I would expect that when a setup provides better performance on the same number of nodes, it is also more energy efficient. How these two are related for different architectures would be an interesting question. I have not any numbers on this though.

- Maybe I missed the information, but do you state somewhere how long the individual experiments were and how many you have run for each performance measure to reduce measurement errors?

The information is in: L. 367, "Each of the runs was repeated three times...". L. 383, the 160km experiment ran for five simulated years. L. 454, the 40km ran for one simulated year.

- I am not quite sure whether you present results from the old and the new machine at DKRZ since it is useful (2 generations of machines), or just since the machine has changed in the meanwhile. Both would be OK but maybe you can justify somewhere.

The first experiments with 160km were performed on the old machine, as the new one was not available yet. When the new machine became available I thought it was a good opportunity to compare the impact of concurrency on the two machines. So it is a bit of both.

Minor points:

L6-7: "The novel…" This sentence is a bit odd and "function parallel technique" is unclear at this stage.

I added a clarification as follows, although the exact meaning has to be developed in the analysis.

"The novel aspect is that component concurrency is a function parallel technique, that is it decomposes the algorithmic space, while these parallelization methods are data parallel techniques, they decompose the data space."

L41: "…cannot efficiently scale" That is quite a statement. Can you provide a reference?

I have slightly changed the sentence to: "In the last years it has become apparent that domain decomposition methods alone cannot efficiently scale...". I did not find a reference on this, and this is partly why I spent a bit of time discussing in this paper the behavior of domain decomposition scaling. While a few years ago there was still some skepticism of the usefulness of shared memory parallelization, like OpenMP, currently I believe all major ESMs employ some form of shared memory parallelization. In other words, the community has voted by their code, and in this sense I consider it to "become apparent".

Figure 1: "sea-land mask is in color" ?

There is one layer of land triangles that serve as boundary to ocean. I removed this detail from the legend as it is not easy to see it.

L113: Better say "local grid-spacings of 600m"

Changed.

L158: Maybe it is just me, but I am not sure what a "trophic level" is.

Trophic levels in general refer to the levels of the food pyramid. In this sentence they refer to the level 1 primary production of organic matter (phytoplankton), and the level 2 organic matter metabolism (zooplankton).

L176: "and may only need… components" Is unclear and should be revised.

We have further clarified it as follows: "The components are algorithmically independent, and may only need to receive input data from other components once in each timestep. For example, the ocean model in a coupled setup requires the atmosphere surface fluxes at the start of each timestep, and then can proceed independently from the atmosphere model."

L190: "in of"

Corrected

You could cite work of ESiWACE from a couple of years ago (but no worries if you disagree): https://zenodo.org/record/1453858#.Y0KPyi8Rp9c

Thanks for pointing this out, I was not aware of this work. I have included a citation in the introduction.

L319: I do not understand this sentence and it should be rephrased.

Changed to: "The second order method flux calculations are based on compatible reconstructions..."

L349: Is there a performance hit when guaranteeing bit-reproducible results? I assume this only holds if the concurrent parts are running on the same hardware?

Bit identical results are obtained on the same hardware when the ocean runs on the same amount of mpi process and the same OpenMP threads (independently of how many procs HAMOCC uses). This has no impact on the performance (optimization flags are set to -O3).

L377: "No such effort…" I think this is an understatement. Maybe say that those parameters have not been optimised?

I think the expression is accurate, the number of mpi procs was predefined, and no calculation took place for improving shared memory balancing.

L410: si -> is

Fixed

L474: "does not apply". Well, I guess this is always only an approximation. -> "is a bad approximation due to the unusual super-linear…"

I changed it to "We note that the assumptions for formula 2 do not apply to this experiment." In this case we have a component with increasing cost as a function of nodes, which is not accounted in this formula. See also line 476: "This cost is significant, and is not been accounted in formula 2."

L513: remove "though"

Done.

---

## Author Comment (AC2)

Thank you for reviewing the paper and for your comments. I have adjusted the revised version of the paper accordingly, and here I provide the answers (in black) to your comments (in blue).

The paper describes the effect of executing concurrently different components of an ESM in terms of performance: ICON-O + HAMMOC. Each of the components are implemented as distributed memory programs (MPI) with multithreaded inter-process execution (OpenMP). A mathematical model of the execution describes the trade-offs of running the components sequentially or in parallel, and gives an intuition on what are the constraints in the expected performance improvements. Performance evaluation is done on two model resolutions and two computer architectures, with an interesting analysis of the results that refers back to the model of execution. This gives the results a conceptual link that is too often neglected in the literature. The paper focuses on ICON-O and HAMMOC, that use the same grid. As a paper I have found it very informative and useful. The mathematical model is not very surprising at the end, but it offers a valuable baseline to reason about the results.

Some points for discussion:

- The abstract needs improvement in my opinion. Line 7 mention "function level parallelism". In the computer science literature the term used is typically "task parallelism", and it is opposed to "data parallelism".

I have rephrased the sentence to "component concurrency is a function parallel technique, it decomposes the algorithmic space, while these parallelization methods are data parallel techniques, they decompose the data space."

I understand that the 'function (or functional) parallelism' term is a bit old-fashion, and 'task parallelism' is more commonly used nowadays. Nevertheless, it is a valid term, and I prefer it for expressing the general concept, as 'task parallelism' in some cases is associated with shared memory parallelization paradigms, for example in OpenMP.

- Would it be possible to explain in few words what are the limitations to scalability mentioned in in line 15 about "traditional parallelization techniques". I do not see the logical implication here, more so given that ICON-O and HAMOCC use the same grid. Is it a problem with software structure? That is, would an implementation with even less modularization (more monolithic) avoid this problem?

This whole paragraph refers to the work done in this paper, to further underline it I start the paragraph with "In this work we study the characteristics...". The phrase "traditional parallelization techniques..." indeed needs clarification: I changed this to "data parallelization techniques (domain decomposition and loop-level shared memory parallelization)...". The scaling limitations of these techniques lie on the size of the grid, not the software structure. This is further analyzed in the paper.

Software structure and engineering is another large subject, which also has implications on multi-level parallelism implementation. This would require a study on its own, as it is rather a complex

issue.

- As my main interest is in software architecture and engineering, I think it would be very interesting to me and useful to the community to expand on the implications that the software restructuring has on the code base. For instance it would be interesting to mention the (qualitative) effects of the ability to run sequentially or concurrently in terms of code maintainability and readability.

This is a very large and complex issue. I can only offer a short comment here. In some cases, the quick and dirty approach is much easier. For example, it's easier to add a few lines of code in the ICON-O surface fluxes module in order to add the HAMOCC surface fluxes, accessing both the ocean and the HAMOCC information, instead of doing so in HAMOCC. Modular design and interfacing is harder, but in the long run probably more robust and advantageous in code-development. A very interesting subject which I would like to discuss in the future, this is a study on its own.

- I find the sentence in lines 214-215, about the fact that OpenMP does not incur in communication costs, not precise. It depends what communications we are considering. OpenMp can be quite costly in case of data to be access by different threads for instance. Maybe the sentence should specify that the communication cost refers to extra-node or extra-process communication.

Here I use the term communication in a strict sense as direct communication between parallel processes. I have added a footnote: "Here by communication it is meant direct communication between the parallel tasks. The cost of it can be significant when it takes place through the network." While there are cases that some communication may occur in shared memory parallel regions (for example reduction, or memory locking) these are not the typical case in our codes. Some communication cost occurs when for example flushing the cache after a parallel region, but I put them under the "overhead" umbrella.

- I find the paragraph in line 227 not very convincing, since it is not clear to me what "a high-level implementation" means here. Is it just because it focuses on MPI to transfer data between MPI ranks?

Indeed, the "high-level implementation" needs further discussion. I have rephrased it as follows: "The other aspect of these two options is the implementation. The distributed memory case is independent of the architecture, and can even be applied in hybrid mode. While in principal the same functionality can be achieved using shared memory parallelization, no such standard, to the authors' knowledge, is currently mature enough to be implemented across multiple architectures." For example, the same concurrent implementation can be used the run ICON-O on CPUs and HAMOCC on GPUS (which is one of our current projects).

- In Section 3.2 I think the treatment could be improved by removing the subscript "p" from "a_p" and use instead the letters "A" or "B" to indicate to which component the value refers to. Also the parameter lambda should be introduced more specifically, since it not immediately clear why the same lambda is used in W_B and N_B. It can be deduced, but it becomes clear later in the text. This could be explained earlier. Similarly, the cost of concurrency "C" could also be introduced with an example of what it may include.

The subscript p is used here to indicate the parallel workload in a parallel region, in contrast to the total workload. This is a bit tricky since a parallel region is defined as any parallel region (shared or distributed) enclosed between two syncs, in any component, independently of sequential or concurrent setup. I have added subscripts A,B,AB to identify the cases of concurrent and sequential runs of A and B.

I attempted to clarify here why N_B = lambda N_A is taken, and to indicate the concurrency cost, but then I would only repeat what follows. On the other hand, one of the advantages of not including this information in this paragraph is to demonstrate that increasing the parallel workload (and thus concurrency) is only useful in the context of scaling concerns.

- The "at most linear" scaling seems to actually mean "monotonic", since F'(N)<=0.

The sentence is rephrased as: "We assume that the parallel efficiency is a non increasing function of N, that is F'(N) <= 0.". This is equivalent to S'(N) <= S(N)/N, where S is the parallel speed-up. In this sense I used the term "at most linear scaling", which I dropped, since this is a bit of a vague term.

- In Section 5 the Authors mention that they ran three times each experiment. It could be useful to report on the variability of the execution times in those three runs to justify the use of such small number. If other limiting factors were in place maybe it is worth mentioning.

The three runs were not meant to provide some statistical confidence, but to avoid accounting a worst case run. In these worst cases the runs can be up to two times slower (due to hardware/system malfunction), and would result a clear outlier. A few of the runs were such cases, but they did not affect the overall results, as they were only one of the three runs.

- In Table 1 and 2 the lambdas are greater that 1, while in the mathematical analysis it is assumed to be less than 1.

Yes, in this case L(N, lambda)=F(N(1+lambda))/F(N  lambda). I have added a clarification on this in the experiments section. I have taken lambda < 1 in the analysis because it makes it simpler. On the other hand, I had in mind the scenario that we have a model A (ICON-O) and we add model B (HAMOCC), so I kept this convention, resulting lambda >1 in the experiments. The conclusions do not change.

- The paper focuses on a simulation software with two components. Could a comment be made on the possibility, both in terms of software structure and performance benefit, of applying concurrency within the said components (I guess in "shared memory" style (see end of Section 3.1))?

Indeed, additional coarse-grained concurrency can be implemented within ICON-O, to the sea-ice module for example. Interestingly, this also probably would be better off with a distributed memory implementation, since the sea-ice is a 2D model, running in an effectively smaller grid than the ocean, but on the other hand it has a lot of global reduction operations, which incur high communication cost for large number of MPI processes.

Shared memory task parallelization can also be further applied. For example in the tracer transport, parallelizing over the number of tracers. In paragraph at line 306 we provide a short discussion. Another candidate for shared-memory concurrency is the diagnostics, the cost of which could be significant depending on the setup. Other processes, such as calculating the tides' potential, may provide opportunities for middle-level shared-memory task parallelism.

- Line 259: I would use "assume" instead of "accept"

Done